# Image Enhancement of Maritime Infrared Targets Based on Scene Discrimination

**DOI:** 10.3390/s22155873

**Published:** 2022-08-05

**Authors:** Yingqi Jiang, Lili Dong, Junke Liang

**Affiliations:** School of Information Science & Technology, Dalian Maritime University, Dalian 116026, China

**Keywords:** maritime infrared image enhancement, target enhancement, image layering, guided filtering, texture feature

## Abstract

Infrared image enhancement technology can effectively improve the image quality and enhance the saliency of the target and is a critical component in the marine target search and tracking system. However, the imaging quality of maritime infrared images is easily affected by weather and sea conditions and has low contrast defects and weak target contour information. At the same time, the target is disturbed by different intensities of sea clutter, so the characteristics of the target are also different, which cannot be processed by a single algorithm. Aiming at these problems, the relationship between the directional texture features of the target and the roughness of the sea surface is deeply analyzed. According to the texture roughness of the waves, the image scene is adaptively divided into calm sea surface and rough sea surface. At the same time, through the Gabor filter at a specific frequency and the gradient-based target feature extraction operator proposed in this paper, the clutter suppression and feature fusion strategies are set, and the target feature image of multi-scale fusion in two types of scenes are obtained, which is used as a guide image for guided filtering. The original image is decomposed into a target and a background layer to extract the target features and avoid image distortion. The blurred background around the target contour is extracted by Gaussian filtering based on the potential target region, and the edge blur caused by the heat conduction of the target is eliminated. Finally, an enhanced image is obtained by fusing the target and background layers with appropriate weights. The experimental results show that, compared with the current image enhancement method, the method proposed in this paper can improve the clarity and contrast of images, enhance the detectability of targets in distress, remove sea surface clutter while retaining the natural environment features in the background, and provide more information for target detection and continuous tracking in maritime search and rescue.

## 1. Introduction

The marine infrared image sequence obtained in complex sea conditions is essential in tracking and detecting targets in search and rescue, military, aerospace, and aviation [1]. However, atmospheric radiation and complex sea conditions affect the marine infrared image, resulting in blurred images with low contrast and signal-to-noise ratios [2]. The target is generally far from the infrared imaging device in practical applications. The size of the pixel occupied by the target in the image is small, and it only appears as a faint point. Its shape details are challenging to estimate, which makes the subsequent target detection and localization work difficult to achieve [3]. The ultimate goal of distress location search is to find the target itself and achieve dynamic detection and tracking of the target. It is also necessary to retain the texture information of the background as a prior condition. Improving the quality of the infrared image of the sea surface, enhancing the details of the area of interest, and improving the detectability of the target are the critical steps for the smooth implementation of the follow-up work in various related fields.

The methods of infrared image enhancement can be divided into five categories: methods based on histogram equalization (HE), methods based on mathematical morphology, methods based on Retinex theory, methods for multi-scale detail extraction, and deep learning methods. The first category of histogram equalization algorithm is widely used due to its simple operation and easy implementation. At the same time, researchers have proposed many improved algorithms, such as contrast limited adaptive histogram equalization (CLAHE) [4], dominant orientation-based texture histogram equalization (DOTHE) [5], and adaptive histogram partition and brightness correction approach (AHPBC) [6]. The over-enhancement phenomenon is improved in CLAHE by limiting the distribution of gray levels. In DOTHE, the texture features of the image are considered to reconstruct the histogram. AHPBC includes three stages: segmentation, local detail enhancement, and noise suppression. To enhance contrast and details of interest, Ashiba et al. also proposed merging gamma correction with histogram matching [7]. The characteristic of this kind of algorithm is that it can significantly improve the contrast of the image and increase the dynamic range of the gray level of the infrared image. However, the most apparent flaw is that it is prone to over-enhancement, which causes the small dim target in the image to disappear. In addition, sharpening and segmentation operations are also introduced in the related improved algorithm, enhancing the background and introducing additional noise.

The second category is the method of mathematical morphology [8], which helps infrared target detection by enhancing dim small targets in infrared images [9,10]. Still, the enhancement results of such methods only retain the regional characteristics of the target, which will lead to image distortion. In addition, morphological operator-enhanced images can be constructed according to the features of infrared images [11]. Still, they do not consider the characteristics of maritime infrared images and are not conducive to enhancing details and detecting targets with complex contours.

The third category is that considering the separation of the target and background area, researchers have applied the algorithm of the Retinex model to infrared images, mainly including the single scale Retinex algorithm (SSR) and the multi-scale Retinex algorithm (MSR) algorithm and its improved algorithm [12]. This kind of algorithm mainly extends from the visible image enhancement method. However, the shortcomings of such algorithms are primarily reflected in the fact that the algorithm is based on the characteristics of human vision and is modeled by the principle of visible light imaging, and the theoretical support is insufficient. The environment of the sea surface is complex, so it is difficult to establish a suitable background model to solve the background radiation component accurately.

The fourth category is based on multi-scale and multi-level detail enhancement methods [13]. To fully extract the tiny details in the image, researchers have proposed related methods, such as the multi-scale feature extraction based on saliency of local windows [14], multi-scale sequential toggle operator using opening and closing as primitives [15], and multi-level image enhancement [16]. Voronin, V et al. proposed an algorithm based on multi-scale alpha-rooting processing, which is enhanced by segmenting image blocks [17]. The main purpose of this kind of method is to extract the details in the image as much as possible, enhancing the background noise. At the same time, they need to overcome the problem of image distortion existing in multi-scale detail enhancement.

The fifth category is the image enhancement method based on deep learning and convolutional neural network (CNN) [18,19,20,21,22,23], which usually requires training a large amount of data on the graphics processing unit (GPU) to obtain a deep learning model for image processing. Such methods are highly efficient but have high demands on the quality and quantity of data, which our maritime infrared target images may not meet.

In addition to the above methods, some other infrared image enhancement methods have achieved good results. In infrared pedestrian detection with converted temperature map [24], the image and temperature domains of pedestrian objects in the image are detected by converting the infrared image to a temperature map. This method takes into account the temperature characteristics of pedestrians. It fully extracts the pedestrian target information in the image, but this method only retains part of the image area of the pedestrian target, and the applicable scene is land. In addition to this, images can be enhanced by removing noise. For example, the image is denoised based on the variance-stabilizing transform and the dual-domain filter [25], which can suppress the mixed Poisson–Gaussian noise and preserve the details of the image. In the method based on wavelet coefficient threshold processing [26], the multiplicative noise is converted into additive noise according to the distribution characteristics of the noise, and the wavelet transform coefficients of the transformed infrared image are denoised. Such methods can effectively suppress noise in terrestrial infrared images, but the noise characteristics in land images are different from maritime images. Enhancing the detail layer of interest through multi-image layering has also been widely used. The learning in-place residual homogeneity for single image detail enhancement (LPRHM) [27] algorithm uses the in-place residual homogeneity to extract detail layers and enhance texture features. Wang, ZJ et al. proposed a layered image enhancement method based on improved guided filtering and compressed the high dynamic range of the image through the distribution information of the histogram, preserving details and suppressing noise [28]. However, their image layering method is not applicable in maritime images.

These methods have achieved good results in infrared image enhancement, but their performance and effects have certain limitations in infrared target images applied in maritime search and rescue. Based on the existing methods, the following four difficulties are mainly faced:The existing methods do not consider the particular circumstances of marine rescue. The target is usually embedded in the background of the waves. The characteristics of the targets under different wave clutter intensities are different and cannot be processed by a single method.The information in the maritime target area is relatively weak. While enhancing the detailed information of the target, it is necessary to overcome noise interference.The target size is uncertain, and the larger target contour is more complex; the smaller target is generally a dim point, so the target feature extraction is more complicated.In the marine rescue operation, it is necessary to realize the positioning and continuous tracking of the target. Therefore, essential features in the background need to be preserved to avoid image distortion.

This paper proposes an image enhancement method for maritime infrared targets based on scene discrimination. The main contributions of the image enhancement method proposed in this paper are as follows:1.For the first problem, an in-depth analysis of the relationship between the characteristics of the target and the sea clutter to improve the generalization of the algorithm to the sea environment, according to the texture roughness of the waves in the local sea area, the scene of the image can be adaptively discriminated, the image is divided into a calm sea image with smooth sea surface and rough sea image with large sea clutter.2.For the second and third problems, according to the difference between the target and background texture features in the two types of scenes, the target features in the two types of images are extracted by the imaginary part of the Gabor filter at a specific scale and the gradient-based target feature operator proposed in this paper, respectively, set different clutter suppression and feature fusion strategies, obtain the target feature image of multi-scale fusion and only enhance the target features.3.For the fourth problem, the target feature image is used as the guide image to conduct guided filtering, and the target layer with similar texture to the guide image is extracted from the original image, which solves the image distortion that is prone to multi-scale feature extraction.4.Finally, according to the principle of thermal conduction in infrared imaging, the blurred background around the target contour is extracted by Gaussian filtering based on the potential target area, the blurred background of the target layer is removed by differential operation, and the appropriate weight is used to fuse with the background layer. It retains the natural environment characteristics in the background.

## 2. Local Gradient Saliency and Multi-Directional Texture Features

A single algorithm is not conducive to processing maritime infrared target images with significant differences in target characteristics under different wave clutter backgrounds. This section will analyze the characteristics of calm sea surface images and rough sea images with varying intensities of the wave from the local gradients in typical regions of the image and the multi-directional texture characteristics of the target to provide a theoretical basis for the subsequent image enhancement algorithm and improve the adaptive sturdiness and robustness.

The scene information of the maritime infrared target image is relatively simple, the image has strong self-similarity, and the correlation between each image area is vital. Taking Figure 1 and Figure 2 as examples, this section will select representative local regions of the image for detailed analysis to reflect the overall image characteristics. When the target in the image is large, the contour of the target will be more complex, and when the target is dim, it will appear as a point. In the selected local area, S, TS, TW, and W were denoted as the sky, targets with complex contours, dim point targets, and sea clutter, respectively. For the rough sea images with only dim small targets, select multiple sea clutter areas to compare with the target area, denoted by W1 and W2, respectively. Analyze the characteristics of the maritime infrared image with the target, and compare the difference between the image of the calm sea surface and the rough sea image.

### 2.1. Local Gradient Saliency

To better analyze the texture intensity of the local area in the image, the gradient in the horizontal and vertical directions of the image was summed, and the gradient magnitude of the specific local area was calculated.

Observing Figure 1 and Figure 2, it can be found that the target in the calm sea image is less disturbed by sea clutter, and the edge contour of the target is complex, which has stronger local gradient saliency than the surrounding background. In the rough sea image, the target was interfered by strong wave clutter, and the gradient amplitude of the target and some strong wave clutter was close. It can be seen from the local gradient amplitude distribution diagram in Figure 3 that the strong amplitude values of the target are relatively concentrated and appear as points.

Comparing the gradient amplitudes in the sea clutter area of the calm sea image and the rough sea image, it can be found that the gradient amplitude in the calm sea image changes gently. However, the sea surface area of the rough sea image has richer texture information and more significant fluctuation of the gray value.

### 2.2. Orientation Texture Feature of Target

From Section 2.1, it can be seen that the gradient magnitude of the target has a strong saliency. We will draw the gradient vector image of the typical local area to analyze the directional features of texture information.

As seen in Figure 4, the target contour in the calm sea image has texture ductility in multiple directions, showing strong continuity. However, the gradient vector changes in the background region are disordered.

The rough sea image target is mainly disturbed by strong sea clutter. The sea clutter has ductility in the horizontal direction, and the target is generally a divergent point. Therefore, the gradient images in four directions of horizontal, vertical, $45^{\circ }$, and $135^{\circ }$ can be calculated, and the texture direction characteristics of the target and the background can be analyzed.

As seen from Figure 5 above, the target area has high gradient values in four directions, of which the target area has the strongest saliency in the $135^{\circ }$ direction. However, it lacks the horizontal stripe texture, and better horizontal texture information can be extracted from the vertical gradient figure.

Through the above analysis, the following three main characteristics of maritime infrared target images can be summarized:
1.Compared with the calm sea image, the rough sea image has a more significant fluctuation in the local gradient magnitude of the sea surface area, the sea surface is rougher, and the texture information is richer.2.The target in the calm sea image is less disturbed by the wave clutter, and the contour of the target is also more complicated, which is also the difficulty of enhancement. The contour of the target extends in multiple directions, which can be fully extracted through multi-scale feature extraction.3.The targets in the rough sea images are mostly dim point targets disturbed by the waves. The difficulty of enhancement lies in the suppression of background clutter in the feature extraction. The gradient of the target in the $135^{\circ }$ direction has stronger significance and can better suppress clutter.

## 3. Algorithm Principle

Based on the analysis of image characteristics in the previous section, an image enhancement method for maritime infrared targets based on scene discrimination was proposed. The overall process is shown in Figure 6, divided into two main steps: image scene discrimination and image detail enhancement. In the image scene discrimination, the image was mainly divided into two categories: rough sea images and calm sea images through the local variance of the sea surface area. In the image detail enhancement stage, the Gabor filter and gradient-based target feature extraction operator proposed in this paper, respectively, perform multi-scale feature extraction on two types of images. Different clutter suppression and feature fusion strategies were set to obtain the target feature images. Using this target feature image as the guide image of guided filtering, the original image was decomposed into a target and a background layer. The blurred background of the target was extracted and eliminated by Gaussian filtering based on the potential target region. Finally, the enhanced target and background layer were fused with appropriate weights to reconstruct the final enhanced image. This chapter will describe each step in detail.

Figure 7 is a block diagram of the proposed method. Combined with Figure 6 and Figure 7, the process of the method proposed in this paper can be better understood.

### 3.1. Scene Discrimination

According to the analysis results in Section 2.1 and Section 2.2, the regional characteristics of the target are different in different situations. From the maritime image, it is found that when the wind and waves in the imaging environment are slight, the sea surface is relatively smooth, and the overall feature information in the image is relatively weak. The feature information is mainly concentrated in the target area; on the contrary, when the wind and waves are significant, the texture of the waves will be rough, and the target in the image will be significantly affected by the wave clutter. Therefore, in this section, we will first use the gamma correction method to process the image to improve the overall contrast of the maritime image. Secondly, the local variance of the background area of the sea surface is used to describe the roughness of the sea wave clutter, and the image scene is adaptively discriminated. The image is divided into calm sea images and rough sea images.

Gamma correction is a nonlinear transformation. The transformation equation is to multiply each pixel value in the original image so that there is an exponential relationship between the gray value of the output image and the original image:(1)$$f(I)=I^{\alpha }$$

Equation (1), α is expressed as a power coefficient. Usually, the value of α is between [0, 2]. In this paper, α=1.5 will be selected, which can maintain the visual effect of the image with better brightness and significantly improve the contrast of the image.

According to the principle of gamma correction, each original gray value will obtain a new gray value after transformation, ignoring the merging phenomenon of a few gray values; gamma correction can ensure that the number of valid gray values of the original image does not change. It effectively improves the contrast of the image without changing the relative size of the original gray level and the number of corresponding gray levels to avoid the loss of dim targets and other detailed information in the image.

Figure 8 is a comparison image of the results after contrast adjustment. Figure 8a,c are the original maritime infrared images collected in the actual scene. Figure 8b,d are the enhancement result images after Figure 8a,c, respectively, and the contrast of the infrared image was significantly improved, and the saliency of the target in the image was also has been improved.

According to the analysis results in Section 2.1 and Section 2.2, it can be seen that the gradient of the sea surface area in the rough sea image is violent, and the texture information is richer. The more significant fluctuation of the waves, the more severe change of the gray value. Therefore, the fluctuation degree of the waves can be described by the standard deviation. The upper part of the image generally has part of the sky area, and the bottom part generally has only the sea surface area. To avoid the texture change of the target area affecting the measurement of sea surface roughness, the area of the bottom 50 rows of the image will be selected and divided into five juxtaposed partial blocks of row height 50. At the same time, considering the significant variance of the target area, the maximum value of the five local variances was deemed to be the potential target area, and the maximum value was removed. The remaining four values are summed to obtain the local variance $SD_{W}$ of the sea surface area. The calculation formula is (2) and (3), M, N are the height and width of the infrared image, $mean_{b}$ represents the mean of partial blocks.
(2)$$std=\sqrt{\frac{1}{M\times N}\sum_{i=1}^{M}\sum_{j=1}^{N}\left[I(x,y)-mean_{b}\right]}$$
(3)$$SD_{W}=\sum_{i=1}^{4}\min(std_{i})(i=1,2,3,4)$$

As shown in Figure 9, 100 images of calm sea and 100 images of rough sea were selected, and the local variance of their sea surface area was calculated using Equations (2) and (3), and the analysis results were plotted. $SD_{W}$ of the rough sea surface is usually higher than that of the calm sea surface, and the strength of waves is proportional to the local variance of the sea surface. According to the above analysis results, when $SD_{W}$ is greater than 40, it is the image taken in the scene of the rough sea; otherwise, it is the image taken in the calm sea scene.

### 3.2. Image Detail Enhancement

The key to image enhancement of marine targets lies in the identification and enhancement of the features of the target area. Enhancing the entire image often results in amplifying the noise of the image. According to the analysis results in Section 2.2, the detail features in the image were extracted according to the directional texture characteristics of the target. Then for the case where the tiny details of the target were distorted, the feature image was used as a guide image to guide the filtering of the original image, and the image was divided into background layers. Plus, the target layer, the tiny texture details of the target were extracted into the target layer for detail enhancement.

#### 3.2.1. Calm Sea Images

In calm sea images, the target has multi-directional texture ductility, the contour of the target is more complex, and the target has strong texture features in multiple directions, so this paper used the imaginary part of the Gabor filter from three frequencies and four directions to achieve the multi-scale contour feature extraction of the target. The related formula of the imaginary part algorithm principle of the Gabor filter is:(4)$$G(x,y,f,\theta )=\frac{f^{2}}{\pi \lambda \eta }\exp(-(\frac{f^{2}}{\lambda^{2}}x^{\prime 2}+\frac{f^{2}}{\eta^{2}}y^{\prime 2}))\sin(2\pi fx)$$
(5)$$x^{\prime }=x\cos\theta +y\sin\theta$$
(6)$$y^{\prime }=-x\sin\theta +y\cos\theta$$

Equations (4)–(6), x,y are used as the Gaussian scale in the principal axis direction and the Gaussian scale orthogonal to the principal axis direction. Among them, θ is the modulating plane wave and the rotation angle of the Gaussian central axis, which is along the counterclockwise direction. In this algorithm, 0, $\frac{\pi }{4}$, $\frac{\pi }{2}$ and $\frac{3\pi }{4}$ were selected to realize the edge feature extraction in four directions in the image. The f is the center frequency of the filter. Different center frequencies will make the light and dark texture of the generated Gabor filter different. The closer the edge contour frequency in the image is to the center frequency of the filter, the larger the response. The values of f are 0.1, 0.2, and 0.3. The λ and η are the spatial vertical and horizontal directions of the x axis and y axis. Considering that the contour of the target in the image is roughly elliptical, the values are λ=1 and η=2, respectively. Using θ and f as variables to construct filter banks of different scales and directions, the edge information of different directions and center frequencies of the image can be detected, respectively.

Each feature submap is extracted at a single frequency and direction, so it is not excluded that some noise will be extracted as the target contour. To suppress noise, non-maximum value suppression (NMS) was performed on each feature submap. Two pixels are found in their filtering and opposite directions for each pixel point. If the point is not the maximum value among these points, it is set to zero; otherwise, keep it. As shown in Formula (7), $F^{\prime }{}_{f_{v},\theta_{u}}(x,y)$ is the feature submap after non-maximum suppression.

A suitable fusion mechanism must be adopted to fuse features at multiple directions and frequencies. The common method for the fusion of feature figures is to take the average value of all submaps or the maximum value. The ultimate goal of this paper was to be able to extract the contour information of the target and determine the approximate area in the original image so that this paper will adopt different feature fusion mechanisms for the scale and direction of the image.

In the calm sea image, except for the target, the background has almost no continuous and regularly changing texture. Hence, the bright features extracted from the feature submap are essential details in the target and are generally reflected as large grayscale values at the scale where they are located. Therefore, for the feature submaps on the three frequencies, the gray value maximum operation was used to fuse the target features of the image at multiple frequencies. The direction of the Gabor filter is non-orthogonal, so the features collected in different directions may overlap, so the average value of each direction submap will be taken when the direction features of the image are fused. As shown in Formulas (7)–(9), $F_{f,\theta }(x,y)$ is the final fusion target feature image obtained.
(7)$$F_{f_{v},\theta_{u}}(x,y)=G_{f_{v},\theta_{u}}(x,y)⊗I(x,y),F^{\prime }{}_{f_{v},\theta_{u}}(x,y)=NMS(F_{f_{v},\theta_{u}}(x,y))$$
(8)$$F_{f,\theta_{u}}(x,y)=\underset{v=i}{\max}[F^{\prime }{}_{f_{v},\theta_{u}}(x,y)](i=1,2,3)$$
(9)$$F_{f,\theta }(x,y)=\underset{u=j}{mean}[F^{\prime }{}_{f,\theta_{u}}(x,y)](j=1,2,3,4)$$

Figure 10 is a fusion feature figure obtained from the calm sea surface. To make the features of the contour of the image continuous, the pixels whose gray value is greater than zero are set to 255. Otherwise, they are set to 0. It can be seen from the Figure 10 that the outline of the target can be determined. Still, the extracted fusion features will appear aliasing distortion, and the weak details inside the target cannot be extracted. This is because the Gabor filter is non-orthogonal, and the same feature may be extracted multiple times. Given this, this section uses the target feature image $F_{f,\theta }(x,y)$ as the guide image to conduct guided filtering to solve the distortion phenomenon of the target feature images. Guided filtering is characterized by edge retention. The feature information of the target area is added through the guide image. The guide image represents the position of the gradient change in the image. The grayscale information of the filtering result is similar to the original image, and strong edges similar to the guide image are retained. The rich details inside the target are smoothed out. By performing the difference operation between the original image and the filtering result, the details of the target can be obtained, and the image can be divided into the target layer and the background layer; Figure 11 shows the hierarchical result, and the equation is (10) and (11):(10)$$B(x,y)=Guide(I(x,y),F_{f,\theta }(x,y))$$
(11)T(x,y)=I(x,y)−B(x,y)

After the target is extracted to the detail layer, the outline of the target needs to be enhanced. The maritime infrared image is based on thermal radiation imaging. The target is generally bright with a gray value higher than the background [29]. According to the principle of heat conduction, higher thermal radiation will diffuse to the surrounding, showing a Gaussian distribution, resulting in blurred edge contours of the target. The higher the gray value in the image, it indicates that the area is a potential area of the target, and the phenomenon of edge blurring caused by heat conduction is more serious. In this paper, the mean value $\bar{\mu }$ of the non-zero pixels of the target layer is calculated as the threshold. The calculation formula is shown in (12), $p_{i}$ represents the gray value of the non-zero pixel. and N represents the total number of non-zero pixels.
(12)$$\bar{\mu }=\frac{\sum_{i=1}^{N}p_{i}}{N}$$

When the pixel value is greater than $\bar{\mu }$, the window centered on the pixel is regarded as the potential target region. Otherwise, it is the background region. Gaussian filtering was performed on the target area (the size of the window selected in this paper was 9×9). A degradation coefficient of β was set to represent the degree of heat radiation conduction. The fuzzy background of the target was obtained, and the final target layer enhancement result was obtained by differential operation between the target layer and the fuzzy background image.
(13)$$T_{E}(x,y)=(F(x,y)-\beta \times F_{g}(x,y))/(1-\beta )$$

Equation (13), F(x,y) represents the target feature image, $F_{g}(x,y)$ represents the blurred background of the target. Generally, the β value is [0.1~0.9]. In this algorithm, β=1/3. The experimental results show that the effect is the best at this time.
(14)$$I_{EN}(x,y)=w_{1}\times B(x,y)+w_{2}\times T_{E}(x,y)$$

$I_{EN}$ represents the final enhanced infrared image, $w_{1}$ and $w_{2}$ is the fusion weight of the resulting image, and increases the contrast of the image by adjusting them. Usually, the weight is a positive value in the range of [0, 3]. To improve the contrast of the image and enhance the details of the image, $w_{1}$ should be smaller and $w_{2}$ should be larger. In this paper, we set $w_{1}=1$ and $w_{2}=1.25$. Experimental results show that these weights are effective.

#### 3.2.2. Rough Sea Images

According to the analysis results in Section 2.2, the target of the rough sea image has good saliency in the gradient image of $135^{\circ }$, and can sufficiently suppress the wave clutter. Still, it lacks the texture features of the target in the horizontal direction. Therefore, in the rough image, the gradient feature image of $135^{\circ }$ and the vertical direction will be extracted with the Sobel operator as the feature submap of the target, as shown in Formula (15):(15)$$\begin{array}{l} F_{135^{\circ }}=\left|G_{135^{\circ }}⊗I(x,y)\right|,F_{y}=\left|G_{y}⊗I(x,y)\right|,\\ G_{135^{\circ }}=\left[\begin{array}{ccc} -2 & -1 & 0\\ -1 & 0 & 1\\ 0 & 1 & 2 \end{array}\right],G_{y}=\left[\begin{array}{ccc} -1 & -2 & -1\\ 0 & 0 & 0\\ 1 & 2 & 1 \end{array}\right] \end{array}$$
(16)$$F_{L}=\sqrt{F_{x}{}^{2}+F_{y}{}^{2}}$$
(17)$$F_{135^{\circ }}^{\prime }=NMS(F_{135^{\circ }}),F_{y}{}^{\prime }=NMS(F_{y})$$
(18)$$F_{G}={\left(F^{\prime }{}_{135^{\circ }}\right)}^{2}\times F_{y}{}^{\prime }$$

As shown in Formula (16), the traditional calculation formula of gradient amplitude is obtained by the gradient amplitudes in the horizontal and vertical directions. After analysis, it was found that it cannot extract the characteristics of the target very well [30]. To solve this problem, we propose a new target feature extraction operator $F_{G}$, which is obtained by multiplying $F^{\prime }{{}_{135^{\circ }}}^{2}$ and $F_{y}{}^{\prime }$. Compared with the traditional gradient magnitude operator, each feature submap performs a non-maximum value suppression, the target saliency in the $135^{\circ }$ direction is amplified, and a certain horizontal texture feature is added. It can be expressed by Formula (18) and $F_{G}$ can also be understood as the target feature image T(x,y) in the rough sea image.

The purpose of image enhancement in this paper is to improve the efficiency of subsequent detection and tracking of marine distress targets. For the rough sea images, we hope to enhance the saliency of the target while retaining the critical texture information of the waves in the background, which is helpful for judgment of the natural environment. A fusion strategy of weighted addition of the target layer and the original image is designed, so Equation (13) can be changed to Equation (19), therefore, I(x,y) is represented as the original image, and the weight settings remain $w_{1}=1$ and $w_{2}=1.25$.
(19)$$I_{EN}(x,y)=w_{1}\times I(x,y)+w_{2}\times T_{E}(x,y)$$

## 4. Experimental Analysis and Results

To verify the effectiveness and robustness of the proposed method, eight infrared images of marine scenes were selected, and the proposed method was compared with the other five methods. Firstly, the experimental data, evaluation indicators, and comparison methods were introduced, followed by an ablation study of the critical steps in the proposed method. Finally, the qualitative and quantitative aspects were compared and analyzed, and the experimental running time of each method was compared. All experiments were conducted on a PC-Windows 10 platform with Intel (R) Core (TM) i5-10210U CPU @ 1.60GHz processor, 16 GB RAM, and the code was implemented in MATLAB R2016a software. The experimental images in this paper were all collected at sea. Figure 12 shows the experimental site and imaging equipment for collecting maritime infrared target images.

### 4.1. Experiment Settings

#### 4.1.1. Experimental Data

This paper uses an infrared imaging system to collect a series of infrared images under different sea conditions and the image size was 640×512, and selects eight scenes with different characteristics. Each scene contains targets, and each infrared image is shown in Figure 13.

The eight images shown in Figure 13, respectively, contain 1–4 targets of different sizes, among which (a–d) of the images are in the calm sea scene, and (e–h) are in the rough sea scene.

#### 4.1.2. Evaluation Metrics

The experimental images in this paper are maritime infrared target images. The characteristic of the image is that the scene information is relatively simple, and the texture information of interest is mainly concentrated on the target. What is more, the purpose of processing images in this paper is to help search and rescue work in maritime distress. Given this, this paper first selected the linear blur metric (γ) [31,32,33,34] to describe the clarity of the image, which is widely used in evaluating infrared images. Secondly, considering that the sea clutter in the image is the main interference factor affecting the image quality, the background suppression factor (BSF) was selected as the evaluation index of the background clutter suppression effect. Finally, to parametrically measure the detectability of the target in distress, the local signal-to-background ratio (LSBR) was used to measure the saliency and detectability of the target.
(20)$$\begin{array}{l} \gamma (I)=\frac{1}{M\times N}\sum_{x=1}^{M}\sum_{y=1}^{N}\min\{p_{xy},(1-p_{xy})\},\\ p_{xy}=\sin[\frac{\pi }{2}(1-\frac{I(x,y)}{L-1})] \end{array}$$

In Equation (20), M×N the infrared image size, I(x,y) is the gray pixel value of (x,y), and L is the maximum grayscale value of I. The smaller the value of γ, the better the performance of the algorithm, the higher the clarity of the image, and the clearer the target texture details in the image.

BSF is usually used to represent the residual degree of background clutters and noise [35,36,37], which can reflect the prominence of the target before and after image processing and reflect the contrast of the maritime infrared target image. The BSF is defined as:(21)$$BSF=\sigma /\sigma_{EN}$$

Equation (21), where σ and $\sigma_{EN}$ denote the standard deviations of gray values in the original and enhanced images. The targets are easily submerged in the cluttered background for maritime images with targets. The larger the variance, the greater the clutter noise in the image, and the more dim the target; the smaller the variance, the better the suppression of clutter in the image. The smoother the image, and the target more prominent. Therefore, the better the performance of the algorithm, the BSF larger, the better the clutter suppression effect, and the stronger the contrast between the target and the background.

The local signal-to-background ratio (LSBR) [33,38,39,40] can characterize the target detection ability of maritime target images. In the maritime infrared target image, the target area is small, and the target gray value in the image is usually larger than that of the background. Therefore, the target in the image is easily misjudged as noise, so SNR is not suitable for judging the local contrast of the target. It can be seen from the maritime target images that the contrast of the target in the image and the ability of target detection are based on the signal strength of the target and the surrounding background, so LSBR is more suitable for characterizing the target detection ability of each algorithm.
(22)$${\mathrm{LSBR}=10\log}_{2}\{\frac{1}{\sigma_{B}^{2}}\sum_{m=-w/2}^{w/2}\sum_{n=-h/2}^{h/2}\left|I(x-m,y-n-\mu_{B})\right|\}$$

In Equation (22), $\sigma_{B}^{2}$ and $\mu_{B}$ represent the variance and mean of the local area of the target, respectively, (x,y) represents the center of the target, w and h represent the width and height of the local window centered on the target. The larger the LSBR, the stronger the detectability of the target. In the case of multiple targets in the image in this paper, the mean value of the LSBR of multiple targets will be calculated. Considering that the size of the targets is inconsistent, the calculation of this value must ensure that the target is embedded in the local window, so the size of the local window is 30×50 or 50×120.

#### 4.1.3. Contrast Method

In this paper, several common infrared image enhancement methods are selected for comparison, including HE, contrast limited adaptive histogram equalization (CLAHE) [4], dominant orientation-based texture histogram equalization (DOTHE) [5], adaptive histogram partition and brightness correction approach (AHPBC) [6], and learning in place residual homogeneity-master (LPRHM) [27].

### 4.2. Ablation Study

The method proposed in this paper has multiple processing steps. Among them, removing the blurred background of the target layer by the Gaussian filtering method based on the potential target area is the most critical step to enhance the details of the target and suppress the clutter background. To verify the effectiveness of this step, this section will conduct an ablation study on the step of removing the blurred background of the target layer. The results of omitting this step are compared with the results of the full proposed method.

Figure 13a,b,e,f were selected for the ablation study, and the comparison results are shown in Figure 14. It can be seen from Figure 14 that the contour details of the target in the image processed by the complete method are clearer, and the contrast of the image is also higher.

In this paper, the purpose of clutter suppression on the target layer was to enhance the details of the target area and suppress the background of sea clutter. Therefore, the values of linear blur metric γ and clutter background suppression factor (BSF) were calculated to verify the effectiveness of this step. Table 1 shows that the γ and BSF of the images processed by the complete proposed method shows better performance, which can effectively reduce the blur of the image and suppress the background of sea clutter.

### 4.3. Comparison of Experimental Results

#### 4.3.1. Qualitative Comparison

In Figure 15a–d are four images containing targets in the calm sea. The HE, CLAHE, and DOTHE methods can improve the contrast of the image. However, they have severe the “over-enhancement” phenomenon, leading to worse visual effects. The LPRHM and AHPBC method both enhance the details of the target area. Still, the LPRHM will add a lot of point noise, and the AHPBC method will reduce the contrast between the target and the background in the image. The method proposed in this paper considers both detail enhancement and contrast enhancement and achieves a good balance between target region enhancement and background suppression.

Figure 15a contains two high-brightness medium targets and two dim point targets close to the sea line. In the HE and DOTHE enhancement results, the faint target closest to the sea line disappeared due to “over-enhancement’”. The LPRHM method improved ship details. However, the faint target is overwhelmed by the added noise. The proposed method performs well, enhancing potential targets and improving the contrast.

In Figure 15b, the HE and DOTHE methods make the distant faint targets disappear. The HE, CLAHE, and DOTHE methods make the fuzzy details inside the ships of large targets disappear, introducing a lot of clutter noise; the horizontal and vertical stripe noise is very obvious. The proposed method enhances dim targets and improves contrast.

In Figure 15c, the HE and DOTHE methods make half of the sea surface seriously distorted, and the sea surface becomes white. The enhancement result of AHPBC makes the sea surface area seriously white, and the contrast of the image is worse than the original image. The proposed method enhances the saliency of the three targets.

In Figure 15d, the HE, CLAHE, and DOTHE methods enhance the contrast between the target and the background but smooth the details inside the target. LPRHM sharpens the outline of the target but also enhances the background. AHPBC hardly changes the image. The proposed method has good performance in improving the internal details and contrast of targets.

In Figure 16e–h there are the four images containing the target in the rough sea image. The HE, CLAHE, and DOTHE methods have the “over-enhancement” phenomenon, resulting in an uneven overall image. LPRHM and AHPBC have a certain enhancement effect on the details of the image but also enhance the background clutter. The proposed method in this paper, while retaining the texture of the waves, soothes the clutter of the sea surface background, dramatically improves the contrast between the sea surface and the target, and significantly improves the detectability of the target.

In Figure 16e, the LPRHM method enhances the details, but the target is weakened. Meanwhile, the point noise produced in the image has a strong similarity with the target, resulting in the target submerged in the background. The AHPBC method has certain denoising, and the background clutter of the image is suppressed. However, the contrast of the image is reduced, and the enhancement effect on the target is not good. The contrast is greatly improved in the HE, CLAHE, and DOTHE methods, and the targets in both the DOTHE and HE methods are submerged in the background due to “over-enhancement”. The proposed method suppresses the noise around the target well and improves the contrast of the image.

In Figure 16f, the wind and waves are enormous in the environment, and the upper left corner is a faint target. The LPRHM method added a lot of extra noise to the image. The AHPBC method smooths the overall image but does not enhance the target. In HE, CLAHE, and DOTHE methods, the target become salient, but the shadow area of the ocean waves is increased, and the streak noise in the image is also enhanced. The proposed method enhances the target and has better visual effects.

In Figure 16g, the target is located near the sea line. The LPRHM method also causes the target to be interfered with by noise, confusing the target and point noise. In the AHPBC method, the sea surface becomes uneven, and the brightness of the sky part increases to make the target faint. In the HE, CLAHE, and DOTHE methods, the sea surface area is inhomogeneity, and the saliency of the target is also weakened. The method proposed can maintain the natural properties of the image and the overall uniformity of the image while improving the contrast of the image.

In Figure 16h, the HE, CLAHE, and DOTHE methods enhance the image contrast but weaken the dim target. The LPRHM method to enhance the target also produces noise. The AHPBC method improves contrast, but the image becomes uneven. The method proposed in this paper enhances the saliency of the target, and the image is clear and natural. Meanwhile, the method proposed in this paper has a good enhancement effect on the targets located near the sea line in Figure 16e,g,h.

#### 4.3.2. Quantitative Comparison

The method proposed in this paper is compared with the five methods of LPRHM, AHPBC, DOTHE, HE, and CLAHE through the metrics of LSBR, BSF, and CLAHE. The first row of each scene is the data value of the original image, and the second row to the fifth row are compared method. The last line is the method proposed in this paper, and the values with better indicators have been marked in bold. BSF needs to calculate the ratio of the variance of the original image and the enhanced result. The original image cannot calculate this parameter so it will be ignored. The calculated metrics results are shown in Table 2 and Table 3:

γ: The linear blur metric of the proposed method is the lowest, which can improve the clarity of the image and suppress clutter. The proposed method can reduce the blurriness to less than half of the original image. In (a) from Table 2, LPRHM, AHPBC, DOTHE, and CLAHE all improve the blurriness of the image. In (c) from Table 2, AHPBC’s method even increases the blurriness of the image up to four times that of the source image.

BSF: The BSF of the proposed methods are all greater than 1, but all other methods are less than 1. It shows that only the proposed method can effectively suppress the clutter background and improve the contrast of the image.

LSBR: In this metric, AHPBC, DOTHE, CLAHE, and HE all appear lower than the original image. The proposed method and the LPRHM performed better. The LPRHM algorithm has the best metric performance, but the proposed method is slightly inferior to the LPRHM algorithm. As shown in Figure 17, this paper selects the target area to draw the grayscale image of all the methods. All five contrast methods generate noise in the background, with the LPRHM method having the most severe point noise. These point noises are considered dim targets and improve the value of LSBR. The point noise weakens the detectability of the target. In the proposed method, the gray value of the target area is improved, and the background noise around the target is also suppressed, which shows that the detection effect of the target in this method is the best.

γ: In terms of the linear blur metric, the AHPBC method increases the blurriness of the image in (e,f,h). The proposed method can improve the clarity of the original image by more than half and has the best performance.

BSF: The BSF of the methods of LPRHM, DOTHE, CLAHE, and HE are all around 0.6. It means that none of them can suppress the wave clutter. The values of the proposed method are all above 1, which can suppress the clutter noise well.

LSBR: In Figure 16g,h, the index of the proposed method is the best. However, the LSBR of the proposed method in Figure 16e,f are slightly lower than the AHPBC method. Given this, as shown in Figure 18, this paper selects the target area in Figure 16e–h, and draws the grayscale image of all the methods. All contrasting methods enhance the background noise, confusing targets and intense noise. The noise is the strongest in LPRHM and AHPBC, which significantly weakens the saliency of the target. The improvement of the LSBR value reflects the enhancement of the noise. Therefore, the proposed method can avoid noise interference and is the best performer in improving the detectability of the target.

#### 4.3.3. Running Time

Here, we will discuss the actual running time of the proposed method. The average running time of the five compared methods and the method proposed in this paper was calculated. Table 4 shows the average running time of each method.

According to Table 4, it can be found that the proposed method in this paper is only faster than AHPBC. To improve the speed, we analyzed each step of the proposed method and found that the extraction of the target feature image took the longest time, wasting more than 1.2266 s. In this operation, multiple target feature sub-maps need to be extracted. Therefore, our team implemented the proposed method on vs2015, and through GPU parallel acceleration technology, the average running time can reach 0.12 s.

To sum up, the proposed method in this paper can improve the detectability of the target and the contrast of the image to a greater extent than the existing methods and better retain the original image feature so that the image can be visually recognized. The effect is closer to the natural high-definition image.

## 5. Conclusions

This paper proposes an image enhancement method of maritime infrared target based on scene discrimination. Under the different wind and wave conditions, the characteristics of the target and the interference of sea clutter are different. To improve the adaptability of the method in the maritime environment, this paper analyzes the features of the maritime infrared target image. According to the local texture roughness of the image, the maritime image is divided into calm sea image and a rough sea image. According to the texture orientation characteristics of the target and background in different scenes, a gradient-based target feature extraction operator is proposed, and clutter suppression and feature fusion strategies are set to extract the target layer. Then, the clutter background of the target layer is removed by Gaussian filtering based on the potential target area, and the details of the target layer are enhanced. Finally, the enhanced target layer is fused with the background layer to obtain the final enhanced image.

The method was compared with the popular methods. It is worth noting that this method has a good enhancement effect on targets of different scales and improves the complex contour details of large targets. Dim small targets near the antenna can ensure they are not lost and improve their salience. This method not only improves the image clarity and enhances the contrast of the image but also improves the detectability of the target. Visually, a good balance is achieved between the target enhancement of the image and the avoidance of image distortion, the original important feature information in the image is maintained, and the image is closer to a high-quality natural image. Compared with the current algorithms, it has stronger generalization ability and robustness. Therefore, the method can provide higher-quality images for maritime distress search and rescue and more helpful information for target identification and continuous tracking.

The disadvantage of the proposed method is that it needs to extract and fuse multiple target feature sub-maps, So there is a limit to the running time. Therefore, our next step will be to improve the speed of the method, which can achieve real-time effects in the maritime search and rescue system.

## Figures and Tables

**Figure 1 sensors-22-05873-f001:**
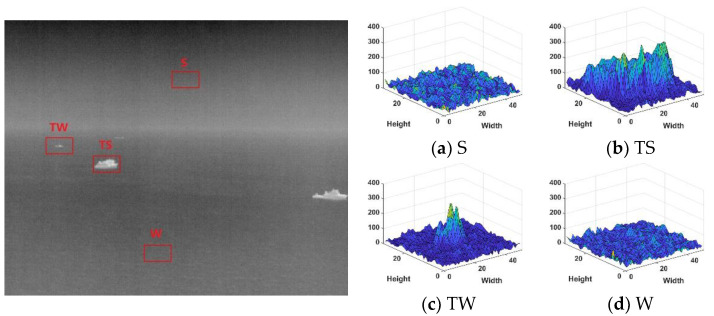
Calm sea image and 3D Fof local gradient magnitude. (**a**) Sky area; (**b**) target with complex contours area; (**c**) dim point target area; (**d**) sea clutter area.

**Figure 2 sensors-22-05873-f002:**
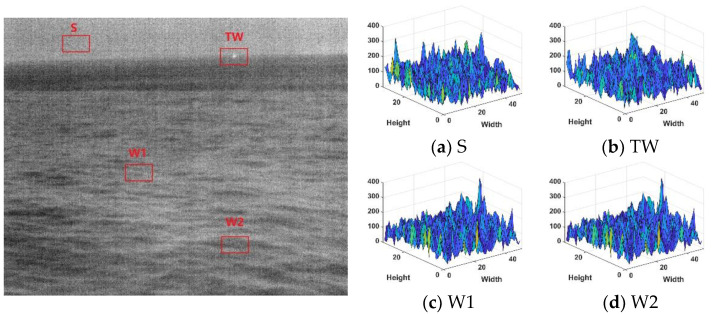
Rough sea image and 3D figure of local gradient amplitude. (**a**) Sky area; (**b**) dim point target area; (**c**) sea clutter area W1; (**d**) sea clutter area W2.

**Figure 3 sensors-22-05873-f003:**
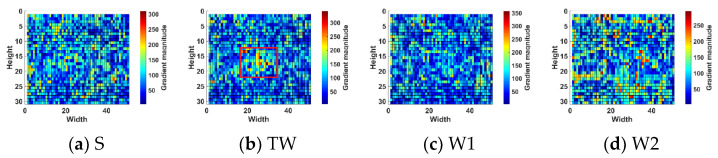
Distribution figure of gradient values. (**a**) Sky area; (**b)** dim point target area; (**c**) sea clutter area W1; (**d**) sea clutter area W2.

**Figure 4 sensors-22-05873-f004:**
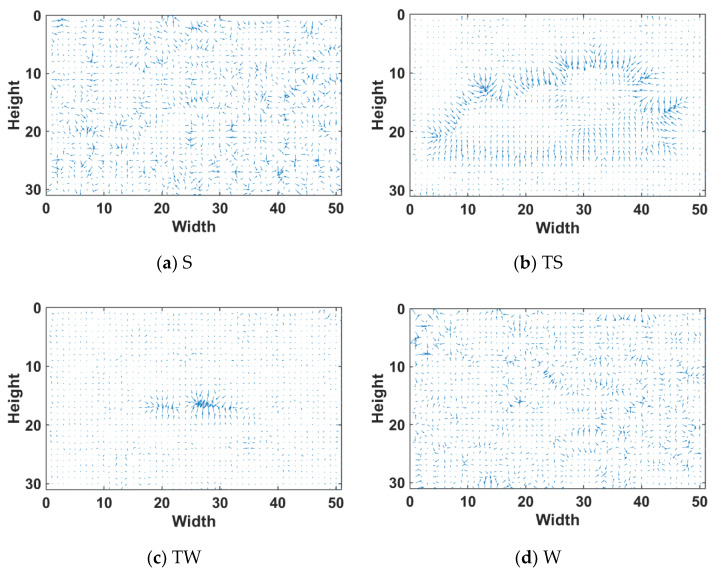
Gradient vector illustration of the local area of the calm sea image. (**a**) Sky area; (**b**) target with complex contours area; (**c**) dim point target area; (**d**) sea clutter area.

**Figure 5 sensors-22-05873-f005:**
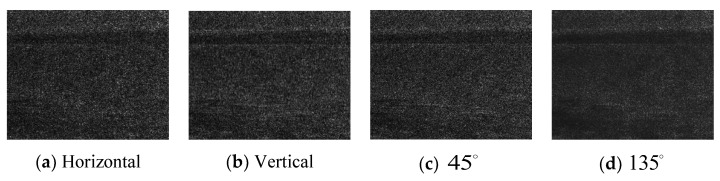
Gradient images of four directions of the rough sea images. (**a**) Horizontal direction; (**b**) vertical direction; (**c**) $45^{\circ }$ direction; (**d**) $135^{\circ }$ direction.

**Figure 6 sensors-22-05873-f006:**
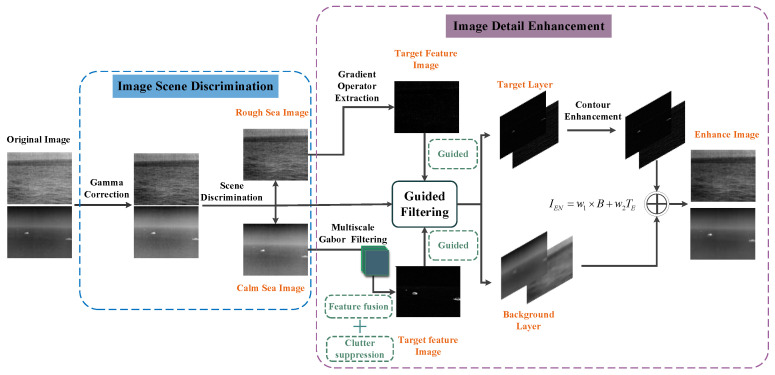
Flowchart of the proposed method.

**Figure 7 sensors-22-05873-f007:**
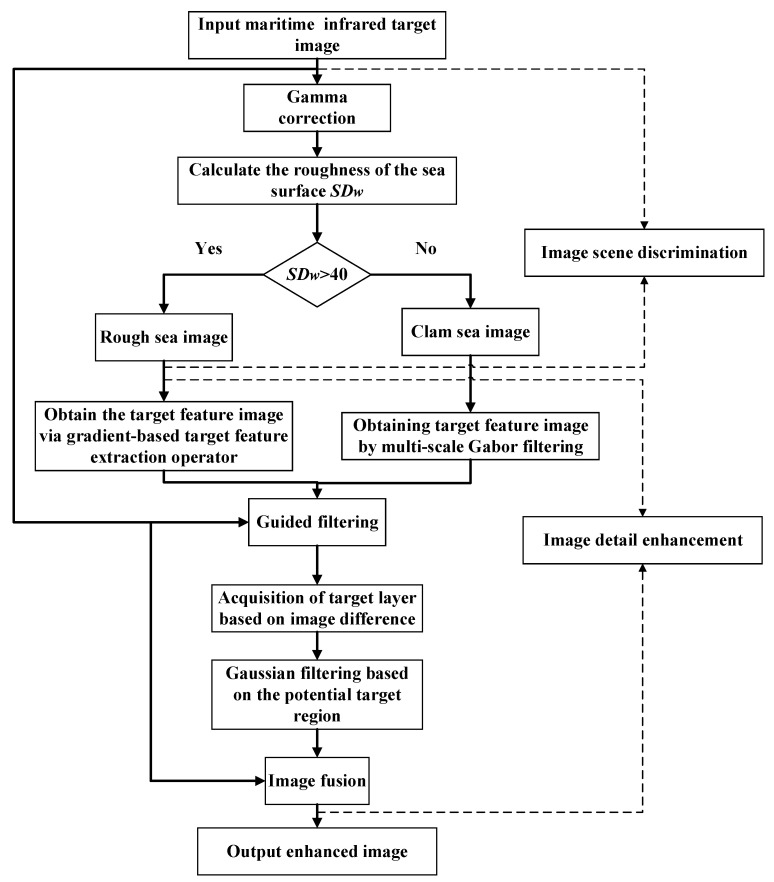
The block diagram of the proposed method.

**Figure 8 sensors-22-05873-f008:**
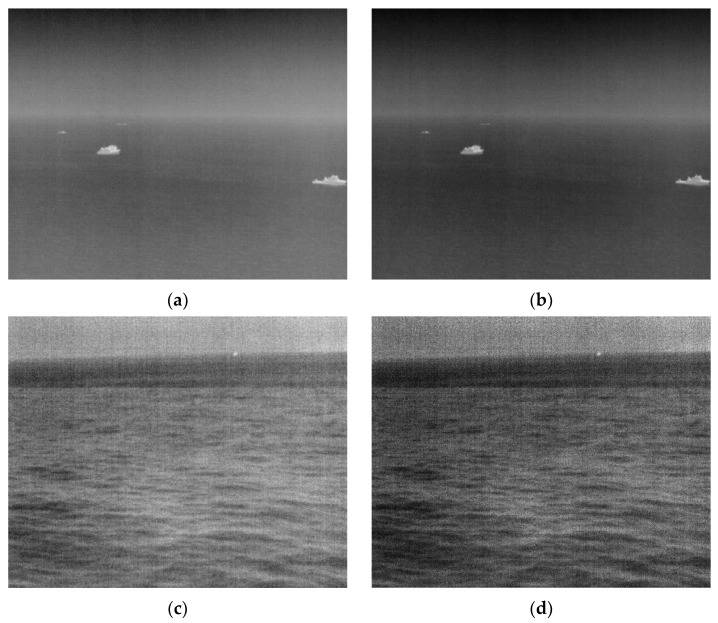
The original image and the result after contrast enhancement. (**a**) Original calm sea image containing two targets with complex contours and two dim targets; (**b**) the result image after the contrast enhancement of (**a**); (**c**) original rough sea image with a dim point target near the sea antenna; (**d**) the result image after the contrast enhancement of (**c**).

**Figure 9 sensors-22-05873-f009:**
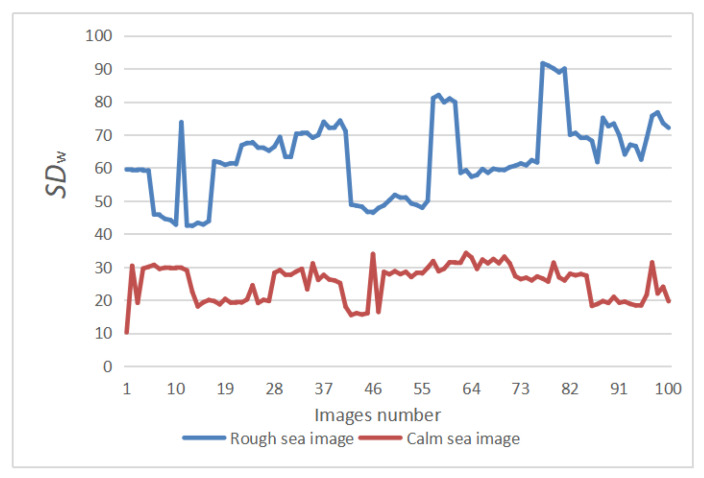
Local variance analysis of sea surface in the maritime image. The red curve represents the variation trend of the local variance of the calm sea image. The blue curve represents the variation trend of the local variance of the rough sea image.

**Figure 10 sensors-22-05873-f010:**
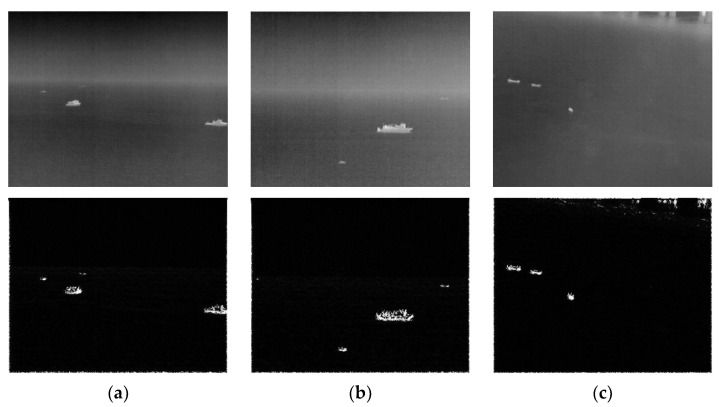
Original image of the calm sea surface and its target feature image. The upper part of (**a**–**c**) is the original image, and the lower part is the target feature image of each image.

**Figure 11 sensors-22-05873-f011:**
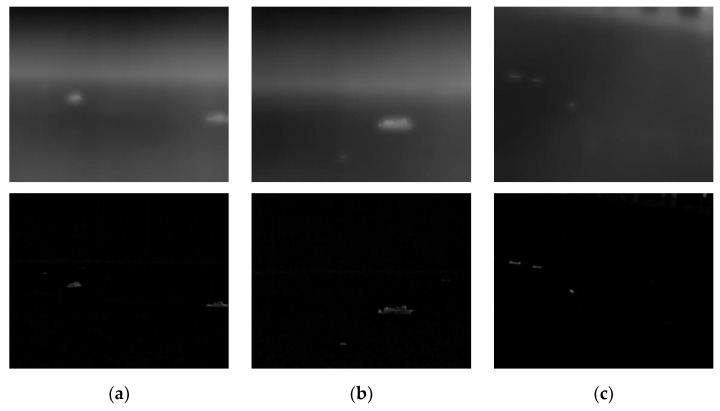
Target and background layers after guided filtering. The upper part of (**a**–**c**) is the background layer and the lower part is the target layer for each image.

**Figure 12 sensors-22-05873-f012:**
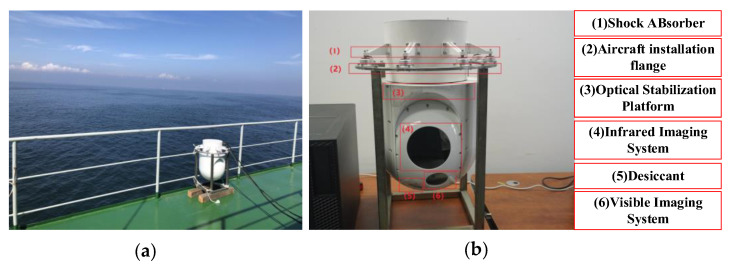
Experimental site and imaging equipment for infrared image acquisition. (**a**) Equipment is collecting infrared images on board around the clock; (**b**) the pod that collects the image, and the important functional blocks of the device are described in the figure.

**Figure 13 sensors-22-05873-f013:**
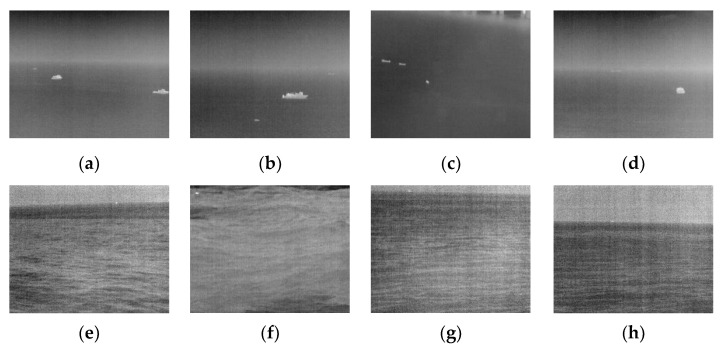
Typical marine infrared target images. (**a**) A calm sea image containing two targets with complex contours and two dim targets; (**b**) a calm sea image with a large ship target and two dim targets, one of which is near the sea line; (**c**) a calm sea image with three small targets; (**d**) a calm sea image with a medium target with a fuzzy dim target near the sea line; (**e**) a rough sea image with a dim point target near the sea antenna; (**f**) a rough sea image with a significant wave fluctuation, and there is a target in the upper left corner; (**g**) a rough sea image with great horizontal wave texture, and there is a target at the top edge of the image; (**h**) a rough sea image with more sky area and a dim target at the sea line.

**Figure 14 sensors-22-05873-f014:**
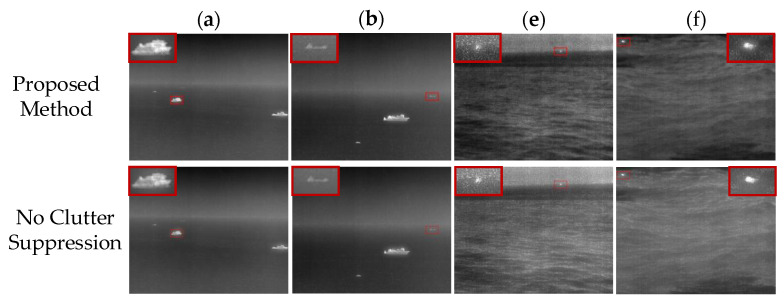
Comparative results of ablation study. The proposed method represents the result image of the complete method proposed in this paper. No clutter suppression indicates the result of no clutter suppression operation. (**a**,**b**,**e**,**f**) correspond to the processing results of the original infrared image in Figure 13, respectively.

**Figure 15 sensors-22-05873-f015:**
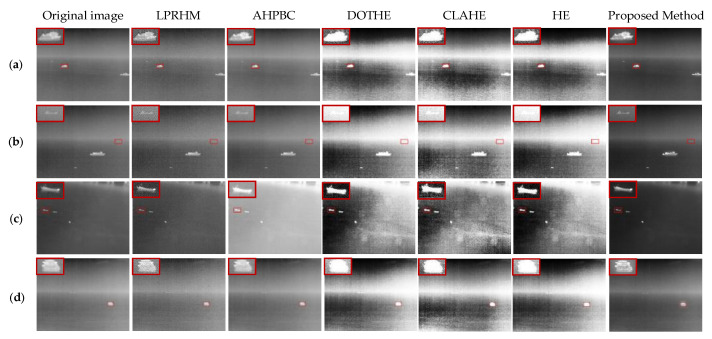
Comparison of the enhancement effect of the calm sea surface. (**a**–**d**) are the experimental results of the calm sea image in Figure 13a–d. The first column is the original image, the second to the sixth are the results of the five comparison methods, and the seventh column is the method proposed in this paper.

**Figure 16 sensors-22-05873-f016:**
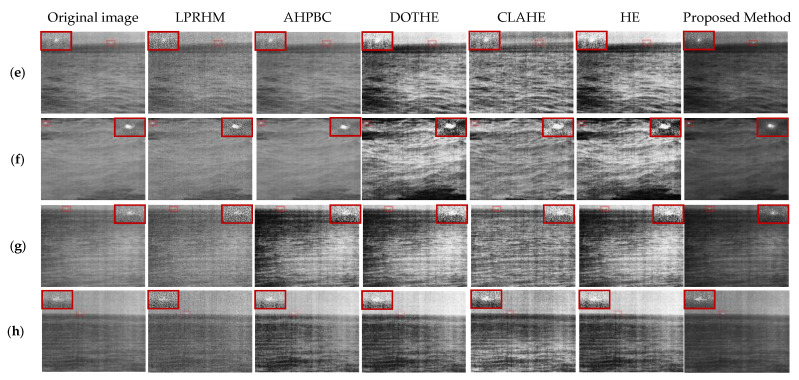
Comparison of image enhancement effects of rough sea images. (**e**–**h**) are the experimental results of the rough sea image in Figure 13e–h. The first column is the original image, the second to the sixth column are the results of the five comparison methods, and the seventh column is the proposed method proposed.

**Figure 17 sensors-22-05873-f017:**
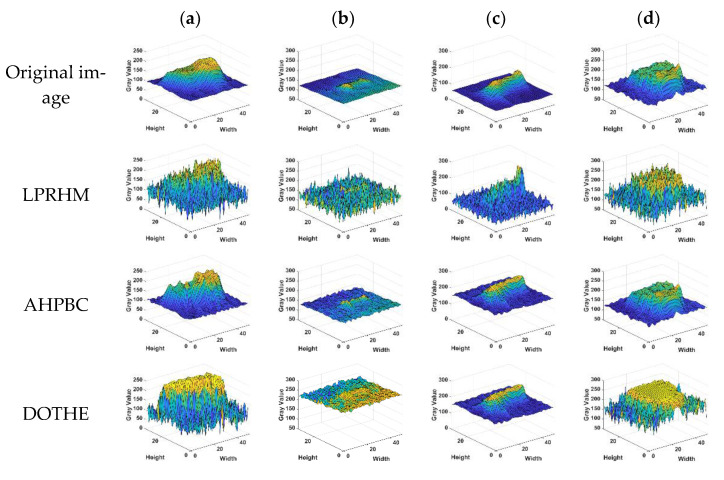
Grayscale image of the target area of calm sea images. (**a**–**d**) are the experimental results of the calm sea image in Figure 15a–d. The first row is the target region of the original image, the second to sixth row are the five contrasting methods, and the last row is the proposed method.

**Figure 18 sensors-22-05873-f018:**
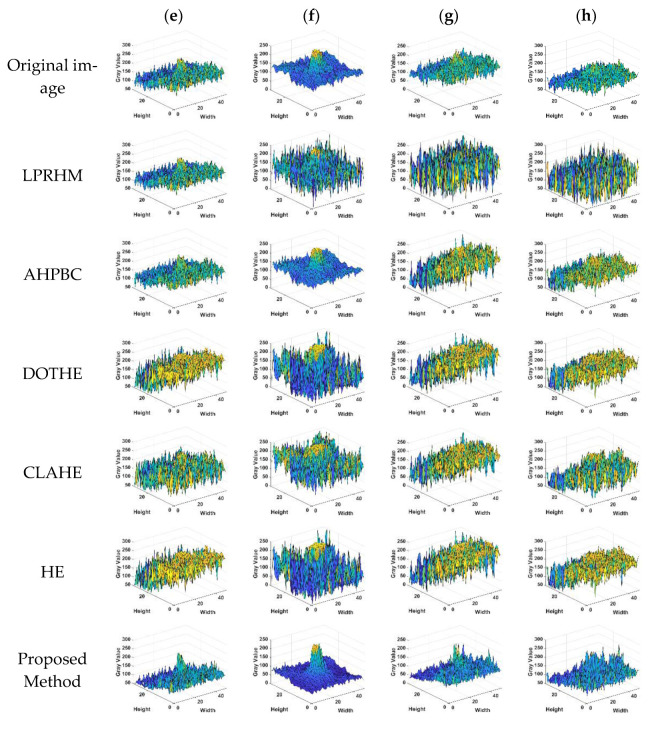
Grayscale image of the target area of the rough sea image. (**e**–**h**) are the experimental results of the rough sea image in Figure 16e–h. The first row is the target area of the original image, the second to sixth rows are the target area of the five contrasting methods, and the last row is the target area of the proposed method.

**Table 1 sensors-22-05873-t001:** Parameter comparison for ablation study. (a,b,e,f) are the corresponding experimental images in Figure 13.

	Methods	γ	BSF
(**a**)	Proposed Method	**0.1951**	**1.0870**
	No clutter suppression	0.2046	0.9978
(**b**)	Proposed Method	**0.1558**	**1.0859**
	No clutter suppression	0.1663	0.9915
(**e**)	Proposed Method	**0.2352**	**1.0296**
	No clutter suppression	0.3928	0.6785
(**f**)	Proposed Method	**0.2150**	**1.1421**
	No clutter suppression	0.3857	1.1203

Bold indicates that this metric has better performance.

**Table 2 sensors-22-05873-t002:** The calm sea image quantitative metrics. (a–d) correspond to the calm sea infrared images in Figure 15a–d, respectively. Different methods process the four images.

	Methods	γ	BSF	LSBR
(**a**)	Original image	0.4003	-	45.0666
	LPRHM	0.4135	0.7405	**50.0929**
	AHPBC	0.4264	0.9964	47.8712
	DOTHE	0.4039	0.3304	46.1790
	CLAHE	0.4030	0.3341	46.9209
	HE	0.3944	0.3257	46.2043
	Proposed Method	**0.1951**	**1.0870**	**48.1291**
(**b**)	Original image	0.3355	-	52.0933
	LPRHM	0.3529	0.7152	**56.2390**
	AHPBC	0.3793	0.9990	53.7169
	DOTHE	0.3985	0.3406	51.9819
	CLAHE	0.4281	0.3693	52.4233
	HE	0.3907	0.3350	51.7469
	Proposed Method	**0.1558**	**1.0859**	**55.8531**
(**c**)	Original image	0.2564	-	43.4092
	LPRHM	0.2659	0.7141	**48.7776**
	AHPBC	0.8101	0.9853	41.9225
	DOTHE	0.4111	0.2412	40.1073
	CLAHE	0.4792	0.2812	41.3876
	HE	0.3985	0.2377	39.8709
	Proposed Method	**0.0991**	**1.0750**	**44.4311**
(**d**)	Original image	0.5182	-	41.6032
	LPRHM	0.5110	0.7649	**47.3336**
	AHPBC	0.5210	0.9959	44.7392
	DOTHE	0.3957	0.4030	44.0986
	CLAHE	0.4201	0.4330	43.1713
	HE	0.3882	0.3966	**45.0032**
	Proposed Method	**0.2941**	**1.0104**	**44.0418**

Bold indicates that this metric has better performance.

**Table 3 sensors-22-05873-t003:** The rough sea image quantitative metrics. (e–h) correspond to the rough sea infrared images in Figure 16e–h, respectively. Different methods process the four images.

	Methods	γ	BSF	LSBR
(**e**)	Original image	0.5610	-	46.9881
	LPRHM	0.4413	0.4940	42.8645
	AHPBC	0.5617	0.9647	**50.5993**
	DOTHE	0.4015	0.4422	42.9489
	CLAHE	0.4596	0.5064	46.2738
	HE	0.3923	0.4357	42.9340
	Proposed Method	**0.2352**	**1.0296**	**49.9856**
(**f**)	Original image	0.6199	-	46.6623
	LPRHM	0.5551	0.4826	46.0825
	AHPBC	0.6291	1.0015	**49.0523**
	DOTHE	0.4009	0.3009	42.6890
	CLAHE	0.5380	0.4382	44.5729
	HE	0.3935	0.2973	42.5791
	Proposed Method	**0.2150**	**1.1421**	**47.6391**
(**g**)	Original image	0.5611	-	47.9276
	LPRHM	0.4363	0.4818	42.6438
	AHPBC	0.4498	0.4557	42.3627
	DOTHE	0.4009	0.4379	42.4675
	CLAHE	0.4477	0.4921	43.7997
	HE	0.3927	0.4313	42.2609
	Proposed Method	**0.2363**	**1.0306**	**50.9625**
(**h**)	Original image	0.4939	-	45.2540
	LPRHM	0.4075	0.5717	42.8781
	AHPBC	0.5191	0.6064	46.5587
	DOTHE	0.4023	0.5326	44.4906
	CLAHE	0.4385	0.5761	44.3757
	HE	0.3950	0.5241	44.3416
	Proposed Method	**0.3534**	**1.0701**	**47.7911**

Bold indicates that this metric has better performance.

**Table 4 sensors-22-05873-t004:** Average running time.

	LPRHM	AHPBC	DOTHE	CLAHE	HE	Proposed Method
Average running time (s)	0.0268	27.7543	0.2352	0.1960	0.0066	1.3298

## Data Availability

Not applicable.

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
