# Peer review of "Image Enhancement of Maritime Infrared Targets Based on Scene Discrimination"

_sensors, 2022, doi:10.3390/s22155873_

Round 1
Reviewer 1 Report
This paper proposes to enhance maritime infrared images based on scene discrimination. Experiments on several image samples have been conducted to validate the effectiveness of the proposed framework. Some detailed comments can be found as follows:
1. Ablation study would be helpful to verify the proposed model, which can show the performance of each key technical component.
2. In figures 13 and 15, all 5 methods can be used for comparisons rather than only one or two methods.
3. In tables 1-2, what is the first “I” method stand for? And some numbers are missing for this method.
4. In the experiments, for the evaluation of enhanced results, why not adopt some mainstream methods, such as SSIM, PSNR, NIQE, etc. Please clarify this point.
5. To provide a better understating for readers, some references about other tasks about infrared images are recommended to be reviewed, which include Infrared pedestrian detection with converted temperature map, Infrared image denoising based on the variance-stabilizing transform and the dual-domain filter, etc.
6. Please further improve the presentation of this paper. For example, the symbols should be unified, i.e. TW and Tw, etc.
Author Response
Dear Reviewer:
We have carefully studied each of your comments and have fully considered and discussed them. According to your comments, we have revised the paper. The attached reply letter is a point-by-point response to the comments, and we wish it to be reconsidered for publication.

Reviewer 2 Report
This paper presents an image enhancement method of maritime infrared targets. However, the writing style is lack of standardization. The comments are as follows:
1. The figures should be improved, such as the labels and the units of coordinate axes are missing, the description of (a)-(f) in Fig.11 are not given.
2. There are so many abbreviations which reduces the readability, and the full names of some abbreviations are not given. The ‘LSBR’ in Eq. (22) and ‘LBSR’ in Table 1 should be same.
3. Some symbols have no explanation,such as ‘c’ in Eq. (1), ‘M’ ,‘N’ and ’mean’ in Eq. (2), etc. Some symbols have different meanings, such as ‘γ’ in Eq. (1), Eq. (4) and Eq. (20), which are easily confused the readers. In Eq.(2) and Eq. (3), ‘i’ have different meanings, and so on.
4. The styles of equations should be unified. For example, there is ‘x’ after ‘w1’ but no ‘x’ after ‘w2’ in Eq. (14) and Eq. (19). The ‘=’ are missing after ‘G135’ and ’Gy’ in Eq. (15). It is improper to use ‘*’ to indicates the dot multiplication in Eq. (18).
5. The symbols (a), (b), (c) in Table 1, and (d),(e),(f) in Table 2 should be explained.
6. The computing times of different methods should be compared in the experiments.
Author Response
Dear Reviewer:
We have carefully studied each of your comments and have fully considered and discussed them. According to your comments, we have revised the paper. The attached reply letter is a point-by-point response to the comments, and we wish it to be reconsidered for publication.
Please see the attachment.

Reviewer 3 Report
-The paper should be interesting ;;;
-it is a good idea to add a block diagram of the proposed research (step by step);;;
-it is a good idea to add more photos of measurements, sensors + arrows/labels what is what (If any);;;
-What is the result of the analysis?;;
-figures should have high quality;;;
-labels of figures should be bigger;;;;
-Figure 1,2, 3, 4 please add labels OX and OY;
-please add photos of the application of the proposed research, 2-3 photos ;;;
-what will society have from the paper?;;
-please compare advantages/disadvantages other approaches;;;
-references should be from the web of science 2020-2022 (50% of all references, 30 references at least);;;
-Conclusion: point out what have you done;;;;
-please add some sentences about future work;;;
Author Response

(The authors gave the same response as above.)

Round 2
Reviewer 1 Report
The authors have addressed all my comments.
Author Response
您的意见对我们很有帮助。非常感谢您的认可。
Reviewer 2 Report
This paper has been highly improved which can be accepted for publication.
Author Response
Your comments is very helpful to us. Thank you very much for your recognition.
Reviewer 3 Report
figures 8-11 please add arrows what is what.
6 selfcitations to sensors it is >10% of selcitations.
It is a good idea to have <10% selfcitations.
Author Response
I have a point-by-point response to the comments, and we wish it to be reconsidered for publication.
Please see the attachment.
